

# Two-hand static and dynamic Arabic sign language recognition using keypoints and shape descriptors with attention-driven feature fusion

Zarnab Kausar[1], Shaheryar Najam[2], Mohammed Alshehri[3],
Yahya AlQahtani[4], Abdulmonem Alshahrani[4], Ahmad Jalal[5,6] and
Jeongmin Park[7]

[1] Department of Biomedical Engineering, Riphah International University, Islamabad, Pakistan
[2] Department of Electrical Engineering, Bahria University, Islamabad, Pakistan
[3] Department of Computer Science, Applied College, King Khalid University, Abha, Saudi Arabia
[4] Department of Informatics and Computer Systems, King Khalid University, Abha, Saudi Arabia
[5] Department of Computing and AI, Air University, Islamabad, Pakistan
[6] Department of Computer Science and Engineering, College of Informatics, Korea University, Seoul, Republic of South Korea
[7] Department of Computer Engineering, Tech University of Korea, Sangidaehak-ro, Siheung-si, Republic of South Korea

Corresponding author
Jeongmin Park,
jmpark@tukorea.ac.kr

## ABSTRACT

Sign language is a vital communication tool for individuals with hearing and speech impairments, yet Arabic Sign Language (ArSL) recognition remains challenging due to signer variability, occlusions, and limited benchmark datasets. To address these challenges, we propose a two-hand static and dynamic gesture recognition system that integrates keypoint-based descriptors (ORB (Oriented FAST and Rotated BRIEF), AKAZE (Accelerated-KAZE), SIFT (Scale-Invariant Feature Transform), and BRISK (Binary Robust Invariant Scalable Keypoints)) with shape-based features (smoothness, convexity, compactness, symmetry) for enhanced gesture discrimination. A distance map-based method is also used to extract fingertip keypoints by identifying local maxima from the hand centroid. An attention-enabled feature fusion strategy effectively combines these diverse features, and a long short-term memory (LSTM) network captures temporal dependencies in dynamic gestures for improved classification. Evaluated on KArSL-100, KArSL-190, and KArSL-502, the proposed system achieved 77.34%, 62.53%, and 47.58% accuracy, respectively, demonstrating its robustness in recognizing both static and dynamic ArSL gestures. These results highlight the effectiveness of combining spatial and temporal features, paving the way for more accurate and inclusive sign language recognition systems.

# INTRODUCTION

Hearing loss is a growing global issue, with the World Health Organization projecting that 2.5 billion people will have some level of hearing impairment by 2050, increasing the need

for sign language communication (*World Health Organization, 2021*). Sign languages like American Sign Language (ASL), Chinese Sign Language (CSL), and Arabic Sign Language (ArSL) have unique linguistic structures and visual vocabularies. ArSL, standardized in 1999, is widely used in Arab countries and media outlets such as Al-Jazeera (*Almufareh et al., 2025*). Sign languages use manual gestures (hand movements) and non-manual cues (facial expressions, posture), with both static (*e.g.*, letters, digits) and dynamic (motion-based) signs. Sign language recognition involves detecting and interpreting visual gestures from images or videos and mapping each sign to its corresponding word or label in the spoken language, thereby facilitating communication with non-signers (*Myagila, Nyambo & Dida, 2025*).

Despite progress in computer vision, ArSL recognition still faces challenges such as signer variability, difficulty capturing non-manual gestures, hand occlusions, inconsistent lighting, and a lack of standardized datasets (*Al-Barham et al., 2023*). These issues hinder system accuracy and require better posture extraction, temporal learning methods, and dataset development. While automated ArSL systems are crucial for inclusive communication, they remain underdeveloped. However, recent advances in vision-based gesture recognition present promising opportunities for improving sign language translation systems (*Alabdullah et al., 2023*).

Hand gestures in sign language are classified as static (fixed postures for letters or numbers) and dynamic (continuous movements for full expressions). Accurate recognition of both is crucial for effective translation. However, challenges like occlusions, hand overlap in two-handed gestures, intra-class variability, and limited research on ArSL continue to affect system reliability (*Al Abdullah, Amoudi & Alghamdi, 2024*).

Recent research has focused on keypoint-based and multimodal approaches to enhance sign language recognition. *Gangwar et al. (2024)* used Speeded-Up Robust Features (SURF) descriptors with a Bag of Visual Words and convolutional neural network (CNN) to classify Indian Sign Language alphabets, performing well on static, single-hand gestures but not addressing dynamic or two-hand signs. *Ferreira, Cardoso & Rebelo (2019)* proposed a multimodal framework using Kinect and Leap Motion sensors to extract complementary features, though it depends on specialized hardware and is not tailored to Arabic Sign Language.

This study proposes a vision-based ArSL recognition system that handles both static and dynamic two-hand gestures without using depth or motion sensors. It combines keypoint-based and shape-based features, with fingertip detection *via* a distance map from the hand's centroid. Features are fused using an attention-enabled module (*Tan et al. 2023*), and long short-term memory (LSTM) networks capture temporal dependencies. The system also uses skin segmentation for consistent hand localization and optical flow to manage occlusions.

The main contributions of this work include:

- Lightweight fingertip detection: We propose a novel distance map-based method to detect fingertips by selecting local maxima from the hand's centroid, enabling efficient and accurate gesture representation.

- Dual-feature representation for gesture recognition: We propose a hybrid approach that combines keypoints-based features and shape descriptors to enhance the recognition of both static and dynamic gestures in ArSL.
- Attention enabled feature fusion: By integrating keypoints-based and shape-based features through a attention enabled feature fusion strategy, our approach improves the system's ability to distinguish visually similar gestures.

The rest of this article is structured as follows: 'Related Work' reviews related work in hand gesture recognition and sign language processing. 'System Methodology' details the proposed methodology, including hand detection, feature extraction, fusion, and classification. 'Experimental Setup' discusses experimental setup and evaluation metrics. 'Results & Discussion' presents results and compares the proposed system with existing approaches. Finally, 'Conclusion and Future Recommendations' concludes the article and outlines future directions.

## RELATED WORK

Hand gesture recognition (HGR) has significantly progressed with advances in deep learning, feature extraction, and multimodal sensor-based methods, becoming essential in human-computer interaction, sign language translation, and smart technologies (*Xie, He & Li, 2018*; *Jalal et al., 2018*). However, challenges like occlusions, viewpoint variations, complex hand articulations, and real-time processing limitations still affect performance (*Chen et al., 2018*; *Liu et al., 2017*). To overcome these issues, researchers have incorporated CNNs, recurrent architectures, and feature fusion techniques to improve robustness and efficiency (*Bhagat, Vishnusai & Rathna, 2019*; *Barbhuiya, Karsh & Jain, 2021*). CNN-based models, especially fine-tuned ones, have achieved high accuracy— 99.82% on ASL alphabets and numerals (*Barbhuiya, Karsh & Jain, 2021*), 99.96% on Indian Sign Language (*Sharma & Singh, 2021*), and 100% on public ASL datasets (*Sharma & Singh, 2021*). Nonetheless, real-time deployment remains challenging due to high computational demands, hand orientation variability, and the need for large-scale annotated datasets (*Ansar et al., 2022*; *Damaneh, Mohanna & Jafari, 2023*).

To overcome these challenges, researchers have explored hybrid models, keypoint-based descriptors, and deep neural networks to improve recognition performance across multiple datasets (*Ansar & Jalal, 2023*; *Mudawi et al., 2024*). Depth information has proven effective in refining gesture segmentation and reducing background interference, while pose-based models have been employed to enhance real-time performance (*Alyami, Luqman & Hammoudeh, 2024*). Capsule networks and transformer-based architectures have also contributed to handling rotation, scaling, and occlusion, making these models more resilient in real-world scenarios (*Shin et al., 2021*; *Sharma & Singh, 2021*).

Recent advances also highlight the importance of temporal modeling and attention mechanisms in improving gesture recognition. *Rahimian et al. (2022)* introduced a Temporal Convolutions-based HGR (TC-HGR) framework incorporating attention modules to reduce computational complexity while maintaining competitive accuracy,

achieving 81.65% on sEMG signals. Building on the TC-HGR framework by *Rahimian et al. (2022)*, *Zhong et al. (2023)* introduced an STGCN-GR model that captures spatial-temporal dependencies in high-density sEMG data, achieving 91.07% accuracy. *Xu et al. (2023)* proposed an SE-CNN architecture using channel-wise attention to enhance feature extraction. *Wang, Zhao & Zhang (2023)* proposed a deep learning approach combining attention mechanisms and transfer learning for electromyographic gesture estimation, improving real-time classification accuracy even with limited training data. These approaches demonstrate the growing importance of selective feature enhancement and temporal dynamics in gesture decoding. *Yu et al. (2023)* developed a CNN-based channel attention model for real-time prosthetic hand control. *Montazerin et al. (2022)* leveraged Vision Transformers (ViT-HGR) to improve classification without heavy data augmentation. These studies underscore the value of temporal dynamics and attention in improving gesture recognition.

Several recent studies have shown promising results using novel approaches. Ansar et al. (2023) developed a CNN-based HGR system using landmark-based features, achieving 93.2% accuracy on the MNIST dataset and 91.6% on the ASL dataset. *Mudawi et al. (2024)* proposed a six-module framework combining SSMD tracking, background modeling, and a 1D CNN, reaching 85.71% accuracy on WLASL and 83.71% on the Indian Sign Language dataset. *Alyami, Luqman & Hammoudeh (2024)* introduced a pose-based transformer model for Arabic Sign Language, achieving 99.74% accuracy in signer-dependent and 68.2% in signer-independent modes on the KArSL-100 dataset. Their model also surpassed existing methods on the LSA64 dataset, with 98.25% and 91.09% accuracy in signer-dependent and -independent settings, respectively.

Incorporating feature fusion and recurrent architectures has also shown strong potential. *Alabdullah et al. (2023)* proposed a markerless dynamic gesture recognition system using joint color cloud, neural gas, and directional active models as features, processed through an RNN. Their model achieved accuracies of 92.57%, 91.86%, and 91.57% on HaGRI, Egogesture, and Jester datasets respectively, and 90.43% on WLASL—highlighting the efficacy of combining advanced feature extraction with recurrent classifiers.

Dynamic gesture recognition, essential for human-robot interaction, has advanced through sequence-based models like ConvLSTMs and transformers (*Bhagat, Vishnusai & Rathna, 2019*; *Ansar et al., 2022*). *Ansar et al. (2022)* achieved 88.46% accuracy on the IPN Hand dataset and 87.69% on Jester using SSD-CNN and temporal features. *Damaneh, Mohanna & Jafari (2023)* improved accuracy and robustness with CNNs, Gabor filters, and ORB descriptors, achieving up to 99.92% on the Massey and ASL datasets.

Hybrid and multi-modal descriptors have also been explored for real-time performance. *Huang & Yang (2021)* developed a finger-emphasized multi-scale descriptor for RGB-D data, demonstrating robustness to articulations and rigid transformations, while *Jalal et al. (2024)* utilized multi-sensor data fusion (RGB + IMU) and hybrid transfer learning to boost recognition accuracy on human activity recognition (HAR) tasks, with results of 92% on the LARa and 93% on the HWU-USP dataset. These methods emphasize

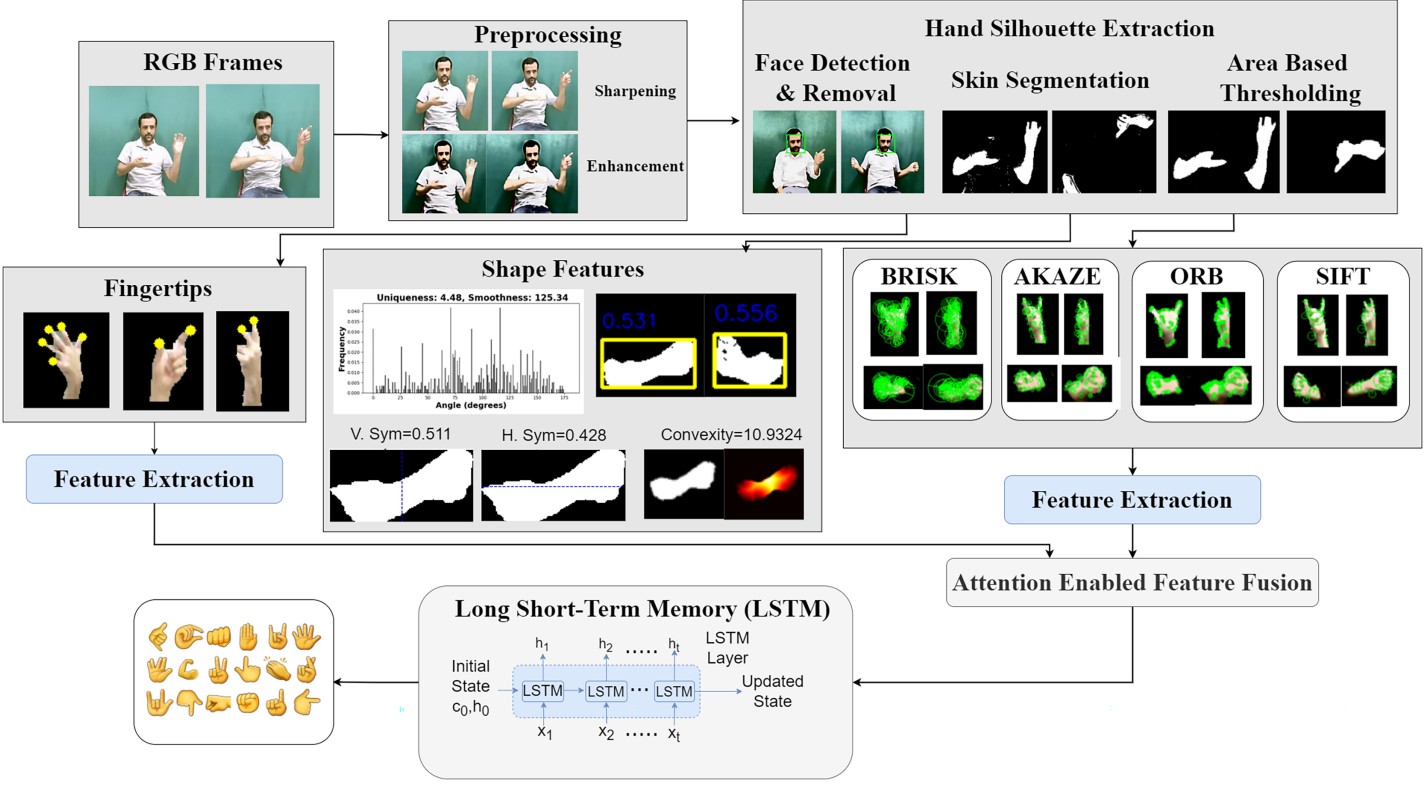

**Figure 1 Comprehensive framework for holistic two hand pose estimation and dynamic motion analysis for gesture recognition.**

the relevance of combining visual cues with sensor fusion and advanced descriptors in achieving real-time recognition.

Despite these advancements, challenges such as real-time efficiency, adaptability to diverse backgrounds, and generalization across different sign languages remain key concerns. The integration of multiple feature extraction techniques and feature fusion strategies, as employed in this work, further enhances recognition accuracy and robustness, making gesture recognition systems more adaptable to real-world scenarios.

## SYSTEM METHODOLOGY

We propose a two-hand Arabic Sign Language recognition system that integrates keypoints and shape features with an attention-driven feature fusion strategy followed by LSTM based classification as shown in Fig. 1.

### Preprocessing steps

The first step in the proposed system is preprocessing the images to enhance quality, remove noise, and improve feature extraction accuracy using three key steps: denoising, sharpening, and brightness & contrast enhancement.

Bilateral filtering enhances gesture recognition by smoothing image while preserving edges, using weights based on both spatial and intensity differences as given in Eq. (1).

$$I'^{(x)} = \frac{1}{W_p} \sum_{i \in \Omega} I(i) f_r(\| I(i) - I(x)\|) f_s(\| i - x\|) \tag{1}$$

where $I(x)$ and $I(i)$ are the central and neighboring pixel intensities within window $\Omega$, with Gaussian blur first applied using Eq. (2).

$$G(x,y) = \frac{1}{2\pi\sigma^2} e^{-\frac{x^2+y^2}{2\sigma^2}} \tag{2}$$

where $\sigma$ represents the standard deviation of the Gaussian kernel. Sharpening is achieved by subtracting the blurred image from the original using Eq. (3).

$$I_{sharp} = I_{original} + \alpha(I_{original} - I_{blurred}) \tag{3}$$

where $\sigma$ controls the sharpening intensity. Finally, brightness and contrast are enhanced using histogram equalization (HE), which redistributes intensity values for better detail visibility using Eq. (4).

$$T(r) = \frac{(L-1)}{MN} \sum_{i=0}^{r} h(i) \tag{4}$$

where $r$ is the input intensity level, $L$ is the total number of intensity levels, $MN$ represents the total number of pixels, $h(i)$ is the histogram count of intensity level.

## Hand silhouette extraction

After preprocessing, faces are detected and removed using Haar Cascade, *Vijaya et al. (2024)*, to focus on hand gestures, using Eq. (5).

$$F(x, y, w, h) = \text{detectMultiScale}(I_{gray}, s, n, m) \tag{5}$$

where, $I_{gray}$ is the grayscale image, $s$ is the scaling factor, $n$ is the minimum neighbors required, and $m$ is the minimum face size. The image is then converted to YCbCr to separate luminance (Y) from chrominance (Cb and Cr), improving skin detection using Eqs. (6a), (6b) and (6c).

$$Y = 0.299R + 0.587G + 0.114B \tag{6a}$$
$$Cb = 128 - 0.168736R - 0.331264G + 0.5B \tag{6b}$$
$$Cr = 128 + 0.5R - 0.418688G - 0.081312B. \tag{6c}$$

A skin mask is created by using Cb and Cr ranges from Eq. (7).

$$133 \le Cr \le 173, \quad 77 \le Cb \le 127. \tag{7}$$

Morphological operations (*Kaur et al., 2024*) refine the mask. Dilation, given by Eq. (8), expands the skin regions, and erosion, given by Eq. (9), removes small noise.

$$D(A) = A \oplus B = \{z | (B)_z \cap A \ne \varnothing\} \tag{8}$$
$$E(A) = A \ominus B = \{z | (B)_z \subseteq A\} \tag{9}$$

where $A$ is the skin mask and $B$ is the structuring element.

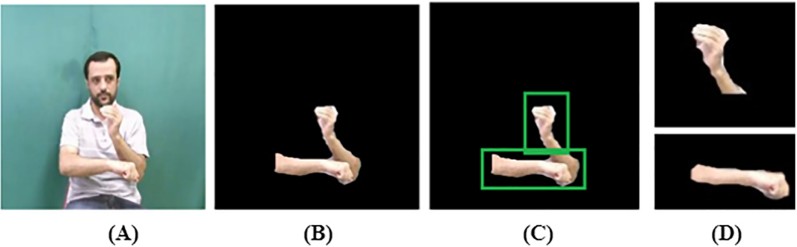

**Figure 2** Silhouette extraction: (A) original image, (B) skin region, (C) occlusion detected, (D) hand separated.               

## Area-based thresholding for hand extraction

Connected component labeling identifies and labels connected regions in a binary image to analyze their properties, using Eq. (10).

$$L(x, y) = \text{Labeling}(I(x, y)) \tag{10}$$

where $L(x, y)$ represents the labeled components. Each component's area is evaluated, and components with an area $A_i \geq 1,500$, are retained while smaller ones are removed. A binary mask is then created to isolate the valid hand components. Further, occlusion occurs when an object is partially hidden by overlapping elements as shown in Fig. 2C. Optical flow detects occlusion regions by analyzing motion discontinuities. Given a flow field with horizontal (u) and vertical (v) components, occlusions can be identified by computing the flow divergence $\nabla \cdot F(x, y)$ as defined in Eq. (11).

$$\nabla \cdot F(x, y) = \frac{\partial u}{\partial x} + \frac{\partial v}{\partial y}. \tag{11}$$

A high divergence magnitude indicates areas where motion abruptly changes, often corresponding to occluded regions. To normalize the occlusion probability, the divergence is scaled using the maximum absolute divergence value across the image using Eq. (12).

$$O(x, y) = 1 - \frac{|\nabla \cdot F(x, y)|}{\max(|\nabla \cdot F|)} \tag{12}$$

where $O(x, y)$ represents the occlusion probability map, where values close to 1 indicate strong occlusions and values near 0 correspond to visible regions. After computing the occlusion probability map $O(x, y)$, the RGB hand region is extracted by focusing on areas with low occlusion, indicating visible parts of the hand. A binary mask $M(x, y)$ is generated using a threshold $\tau$, typically between 0.3 and 0.5, to filter out heavily occluded pixels using Eq. (13).

$$M(x, y) = \begin{cases} 1, & \text{if } O(x, y) < \tau \\ 0, & \text{otherwise} \end{cases}. \tag{13}$$

This mask highlights only the reliable, unoccluded regions. The final RGB hand image $H(x, y)$ is obtained by applying this mask to the original RGB frame $I(x, y)$ using Eq. (14).

$$H(x, y) = I(x, y) \cdot M(x, y). \tag{14}$$

To enhance spatial coherence and remove noise, morphological operations such as closing and dilation are applied to $M(x, y)$ before masking. This process ensures accurate extraction of the visible hand region, even under partial occlusions, enabling reliable downstream gesture or pose analysis.

## 2D hand pose estimation

The segmented hand region is resized to $400 \times 400$ pixels using bilinear interpolation to ensure uniformity and maintain aspect ratio. Keypoints and descriptors are then extracted using Oriented FAST and Rotated BRIEF (ORB), Accelerated KAZE (AKAZE), Scale Invariant Feature Transform (SIFT), Binary Robust Invariant Scalable Keypoints (BRISK) and distance map for robust feature detection in gesture recognition.

### A. ORB

ORB (*Tareen & Saleem, 2018*) is used for feature extraction and keypoint detection from the segmented hand region, identifying gesture landmarks crucial for Sign Language recognition. To balance accuracy and computational cost, 200 keypoints are extracted as shown in Fig. 3A. ORB detects keypoints using a modified FAST algorithm where a pixel p is a keypoint if a contiguous arc of $n$ pixels meets the condition given by Eq. (15).

$$\left|I_x - I_p\right| > t, \quad \forall x \in S \tag{15}$$

where $I_x$ and $I_p$ are the pixel intensities, $S$ is the circular neighborhood of 16 pixels around $p$, $t$ is a threshold for intensity difference. ORB ensures rotation invariance by assigning an orientation using the intensity centroid $C$ given by Eq. (16).

$$C = \left( \frac{\sum xI(x, y)}{\sum I(x, y)}, \frac{\sum yI(x, y)}{\sum I(x, y)} \right) \tag{16}$$

where $I(x, y)$ is the pixel intensity. Using this centroid, ORB assigns an orientation angle $\theta$ to each keypoint by Eq. (17).

$$\theta = \tan^{-1} \left( \frac{\sum yI(x, y)}{\sum xI(x, y)} \right). \tag{17}$$

For each key point in the hand, ORB extracts a binary feature descriptor by comparing the intensity values of pairs of pixels using Eq. (18).

$$f(i) = \begin{cases} 1, & I(x_i) < I(y_j) \\ 0, & \text{otherwise} \end{cases} \tag{18}$$

where $x_i, y_j$ are pixel pairs sampled within a local region. $I(x_i)$ and $I(y_j)$ are their intensity values.

### B. AKAZE

AKAZE (*Tareen & Saleem, 2018*) is a fast, robust feature detection algorithm using nonlinear scale spaces to extract key points from hand components, aiding in tracking finger positioning, shape, and movement by detecting high-contrast regions, edges, and corners. The key points, provided in Fig. 3B, are computed using the Perona-Malik

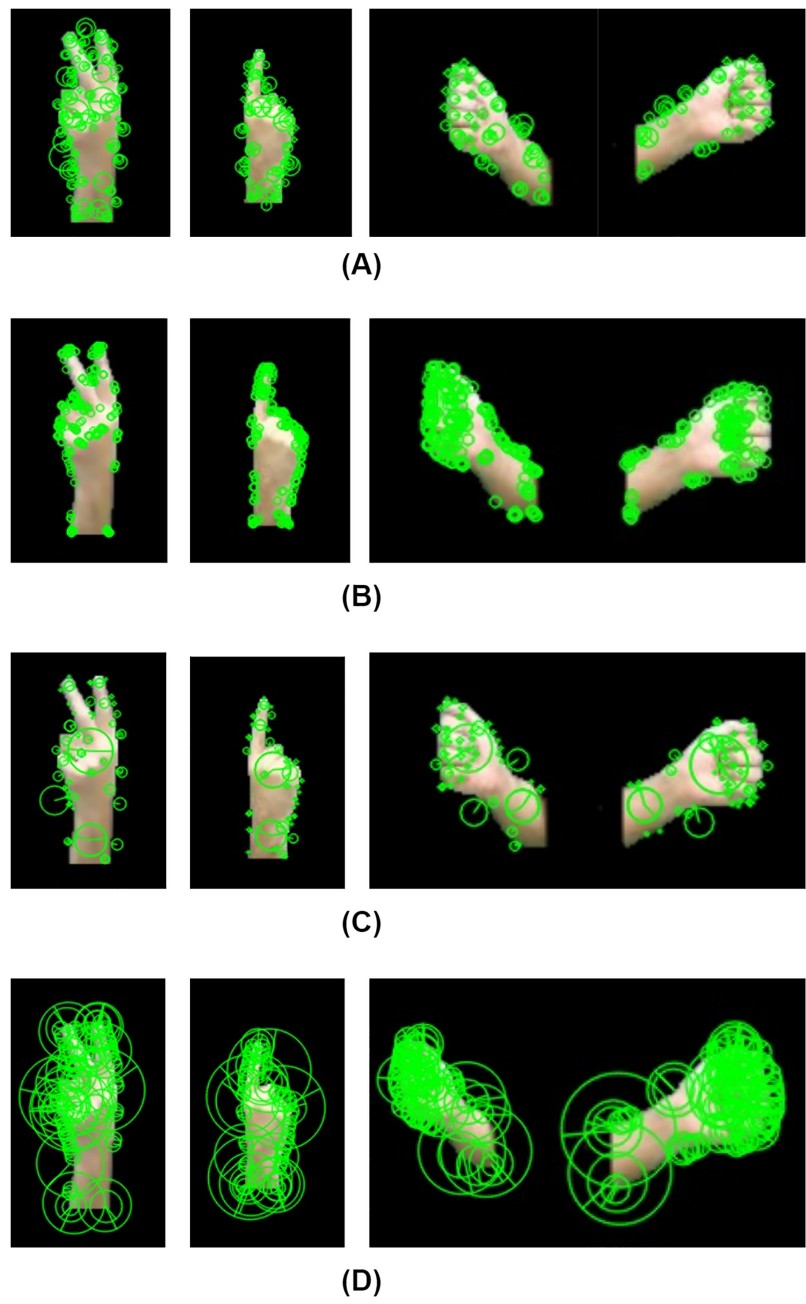

**Figure 3 Keypoints extraction using (A) AKAZE (B) ORB (C) SIFT (D) BRISK various gestures.
Left: "2" Centre: "1" Right: "محامً".**

anisotropic diffusion equation given in Eq. (19), with smoothing controlled by diffusion function given in Eq. (20)

$$\frac{\partial L}{\partial t} = \nabla \cdot (g(\nabla L)\nabla L) \tag{19}$$

$$g(\nabla L) = \frac{1}{1 + \left(\frac{|\nabla L|}{\kappa}\right)^2} \tag{20}$$

where $L$ is the image intensity function, $\nabla L$ represents the image gradient, $\kappa$ is a contrast-sensitive threshold. Each key point is associated with a descriptor that encodes local texture and shape information. The descriptor vector is computed using Binary Robust Independent Elementary Features BRIEF-like descriptors. Given a detected key point $k$ at location $(x_k, y_k)$, the descriptor $D_k$ is constructed as given by Eq. (21).

$$D_k = \left[ I(x_k + \delta x_i, y_k + \delta y_i) - I(x_k + \delta x_j, y_k + \delta y_j) \right]_{i,j=1}^{N} \tag{21}$$

where $I(x, y)$ is the image intensity, $\delta x_i, \delta y_i$ and $\delta x_j, \delta y_j$ are small perturbations around the key point, $N$ is the number of binary comparisons used to form the descriptor. To ensure computational efficiency, top-ranked key points are retained based on their response strength $R_k$, given by Eq. (22).

$$R_k = \sum_{i=1}^{N} D_k(i). \tag{22}$$

These key points represent anatomical landmarks on the hand, enabling estimation of hand orientation, finger spread, and motion patterns.

*C. SIFT*

SIFT (*Tareen & Saleem, 2018*) extracts distinctive, scale- and rotation-invariant keypoints from hands as shown in Fig. 3C. To detect keypoints in hands at different scales, SIFT builds a scale-space representation by progressively blurring the hand region using a Gaussian function by using Eq. (23), where $I(x, y)$ is the processed hand image, $G(x, y, \sigma)$ is a Gaussian kernel given by Eq. (24).

$$L(x, y, \sigma) = G(x, y, \sigma) * I(x, y) \tag{23}$$

$$G(x, y, \sigma) = \frac{1}{2\pi\sigma^2} e^{-\frac{x^2+y^2}{2\sigma^2}}. \tag{24}$$

To efficiently find key points, Difference of Gaussians (DoG) is computed by subtracting two Gaussian-blurred images at different scales given by Eq. (25).

$$D(x, y, \sigma) = L(x, y, k\sigma) - L(x, y, \sigma) \tag{25}$$

where $k$ is a constant multiplier. Keypoints are detected as local extrema in the DoG images by comparing each pixel with its 26 neighbors. Keypoints are refined using Hessian matrix at each location with the response ratio $R$ is computed by Eq. (26).

$$H = \begin{bmatrix} D_{xx} & D_{xy} \\ D_{xy} & D_{yy} \end{bmatrix}, \quad R = \frac{\text{Tr}(H)^2}{\det(H)} \tag{26}$$

where $\text{Tr}(H)$ and $\det(H)$ are calculated by using Eqs. (27) and (28).

$$\text{Tr}(H) = D_{xx} + D_{yy} \tag{27}$$

$$\det(H) = D_{xx}D_{yy} - \left( D_{xy} \right)^2. \tag{28}$$

Each keypoints is assigned a dominant orientation for rotation invariance. The gradient magnitude and direction are computed for each key point's neighborhood using Eqs. (29) and (30) respectively.

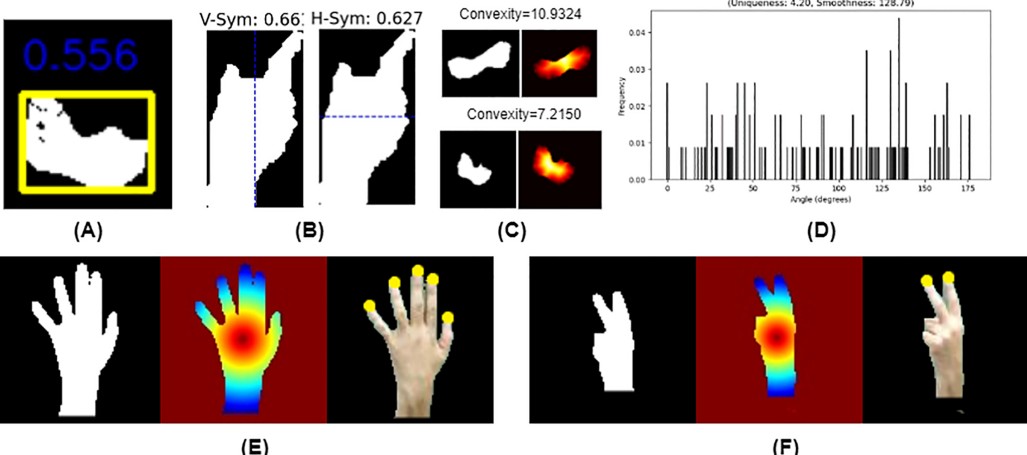

**Figure 4** Shape features: (A) uniqueness and smoothness, (B) convexity, (C) compactness (D) vertical and horizontal symmetry distance based keypoints: (E) "5": binary silhouette (Right) distance map (Middle), fingertips keypoints (Left), (F) "2": binary silhouette (Right), distance map (Middle), fingertips (Left).                               

$$m(x,y) = \sqrt{\begin{array}{l}(L(x+1,y)-L(x-1,y))^2 \\ + (L(x,y+1)-L(x,y-1))^2\end{array}} \tag{29}$$

$$\theta(x,y) = \tan^{-1}\left(\frac{L(x,y+1)-L(x,y-1)}{L(x+1,y)-L(x-1,y)}\right). \tag{30}$$

For dynamic gestures, keypoints from sequential frames are matched using Euclidean distance Outliers are removed using RANSAC (Random Sample Consensus) to eliminate incorrect matches due to background noise.

### D. BRISK

BRISK (*Tareen & Saleem, 2018*) identifies keypoints, illustrated in Fig. 4D, using a scale-space pyramid approach, ensuring robustness to variations in hand size and orientation. It constructs a pyramid of images at different resolutions by iteratively down sampling the original image using Eq. (31).

$$I_s(x,y) = (I_{s-1} * G_{\sigma_s})(x,y) \tag{31}$$

where $G_{\sigma_s}$ is a Gaussian kernel with standard deviation $\sigma_s$. At each scale s, a modified FAST detector identifies corners. A pixel p is considered a corner if a contiguous arc of at least n pixels in a circular neighborhood is significantly brighter or darker than p as given in Eq. (32), where q is a neighboring pixel of p, and t is the intensity threshold.

$$|I(p) - I(q)| > t. \tag{32}$$

After keypoints detection, subpixel interpolation refines their locations for accuracy. BRISK computes a binary descriptor by comparing intensity values at 60 predefined points in circular sampling, given by Eq. (33), resulting in a 512-bit descriptor.

$$f(i,j) = \begin{cases} 1, & I(p_i) < I(p_j) \\ 0, & \text{otherwise} \end{cases}. \tag{33}$$
## SHAPE FEATURES FOR BINARY HAND SILHOUETTES

Binary silhouette of the hand region is analyzed using shape parameters which provide a mathematical representation of the hand's structure, enabling precise differentiation between various hand poses (*Murat, 2024*). Their computation and corresponding formulas are detailed below:

*A. Compactness:*

Compactness is defined as the ratio of the actual area of a detected object (hand or finger) to the bounding box area enclosing it given in Eq. (34) as shown in Fig. 4A.

$$C = \frac{A}{B} \tag{34}$$

where $A$ represents the actual area of the detected hand region (from the binary skin mask) and $B$ is the area of the minimum enclosing bounding box.

*B. Symmetry*

Symmetry is evaluated by comparing the original hand region with its flipped version along a chosen axis (horizontal or vertical). The symmetry score is computed using Eq. (35) and is illustrated in Fig. 4B.

$$S = \frac{O}{\max(A, 1)} \tag{35}$$

where $O$ is the overlapping area between the original hand mask and its flipped counterpart, $A$ is the total area of the hand region in the binary mask.

*C. Global convexity*

Global convexity, provided in Fig. 4C, quantifies the deviation of the hand region from its convex hull by measuring the Euclidean distance between each pixel of the hand silhouette and the nearest convex hull point. The global convexity score is computed using Eq. (36).

$$C_g = \frac{1}{N} \sum_{i=1}^{N} d_i \tag{36}$$

where $N$ is the number of pixels in the region, $d_i$ is the Euclidean distance between the $i^{th}$ pixel of the hand region and the nearest convex hull point.

*D. Uniqueness and smoothness*

Uniqueness quantifies the distinctiveness of gesture by analyzing the distribution of tangent angles along the contour, computed using the Shannon entropy of the angle histogram using Eq. (37) as shown in Fig. 4D.

$$U = -\sum_{i=1}^{N} P_i \log_2 P_i \tag{37}$$

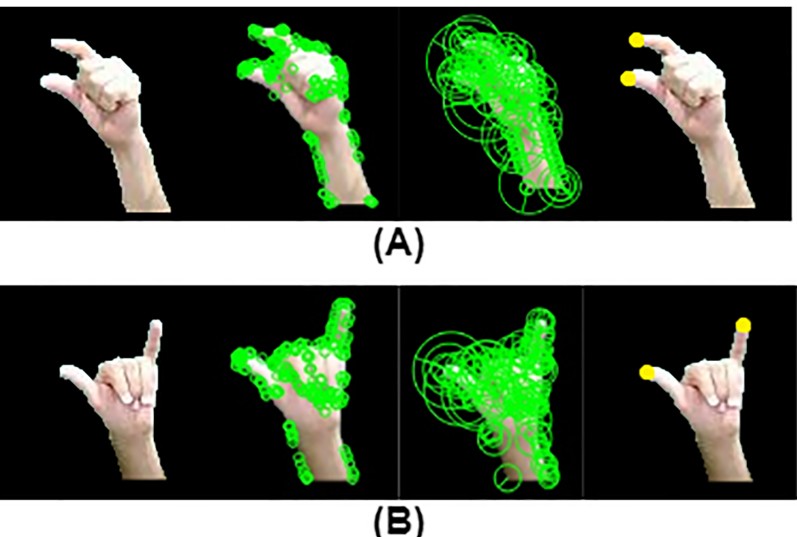

**Figure 5** Results of ORB, BRISK and fingertip locations for fingerspelling gestures: (a) " د": (b) " ي".

where $P_i$ is the probability of an angle falling into the $i^{th}$ segment of the histogram and $N$ is the number of segments. While smoothness evaluates the continuity of a hand gesture by analyzing variations in the contour's tangent angles, provided in Fig. 4D. It is computed using Eq. (38).

$$S_m = \max(\mu_1, \mu_2) \tag{38}$$

where $\mu_1, \mu_2$ are the mean angle values obtained from a Gaussian mixture model (GMM) fitted to the angle histogram.

## Keypoints calculation using distance map

To compute keypoints using distance map, first the centroid of each hand component is computed using image moments using Eq. (39).

$$c_x = \frac{\sum x_i M(x_i, y_i)}{\sum M(x_i, y_i)}, \quad c_y = \frac{\sum y_i M(x_i, y_i)}{\sum M(x_i, y_i)} \tag{39}$$

where $M(x, y)$ is the binary mask of the segmented hand region. A distance transform is applied to calculate the Euclidean distance from each pixel $(x, y)$ in the hand region to the centroid. Local maxima in the distance map *i.e.*, $\max(D(x, y))$ are identified as fingertip locations. A local window-based maximum filter ensures robust fingertip selection. The five most distant points from the centroid are selected as fingertips using Eq. (40) as shown in Figs. 4E and 4F.

$$F = \{(x_i, y_i) \mid i \in [1, 5], D(x_i, y_i) \text{ is top 5 maxima}\}. \tag{40}$$

While some preprocessing techniques may eliminate hand features critical for recognizing fingerspelling, a key component of sign language, our approach preserves the

hand configurations necessary for accurate interpretation. As demonstrated in Fig. 5, this enables reliable recognition of several fingerspelled Arabic alphabet signs.

## Feature extraction from keypoints

We extracted features from keypoints to analyze keypoint motion in gesture recognition. Displacement, velocity, acceleration, angular displacement, angular velocity, and angular acceleration were computed with the corresponding formulas provided by *Murat (2024)*.

## ATTENTION ENABLED EARLY FEATURE FUSION

For each frame, we extracted a total of 6,454 features. This includes eight features per keypoint, comprising displacement in x and y (2), linear velocity (2), linear acceleration (2), angular velocity (1), and angular acceleration (1). With 800 keypoints across four regions (200 each), this yields 6,400 features. Additionally, we computed eight features from six aggregated vectors ($6 \times 8 = 48$) capturing overall movement trends. Lastly, we included six global shape features—horizontal and vertical symmetry, compactness, convexity, smoothness, and uniqueness—bringing the total feature count per frame to 6,454.

Each frame of the input video produces two feature vectors: a skeletal-based vector $s_i \in R^{N_1}$ and a silhouette-based vector $h_i \in R^{N_2}$. These features are concatenated to form the early fused feature vector $f_i \in R^{N_1+N_2}$ as defined in Eq. (41):

$$f_i = s_i \oplus h_i \tag{41}$$

where, $\oplus$ denotes the concatenation operation. For a video sequence of M frames, the complete fused feature matrix $F \in R^{M \times (N_1+N_2)}$ s represented using Eq. (42).

$$\mathbf{F} = [f_1, f_2, \ldots, f_M]. \tag{42}$$

To emphasize the most informative components in the concatenated feature vector $f_i$ a self-attention mechanism is applied. For each fused vector $f_i$, a corresponding query vector $q_i$ and key vector $k_j$ are computed through linear transformations with learned parameters: $q_i = W_q f_i + b_q, \quad k_j = W_k f_j + b_k$ where, $W_q \in R^{d \times (N_1+N_2)}$, $W_k \in R^{d \times (N_1+N_2)}$ $b_q \in R^d$ $b_k \in R^d$ are trainable parameters learned *via* backpropagation during training. The attention score-$e_{ij}$, reflecting the similarity between the query $q_i$ and key $k_j$, is computed using scaled dot-product attention in Eq. (43).

$$e_{ij} = q_i^T \cdot k_j. \tag{43}$$

These raw attention scores are normalized with the softmax function to yield attention weights $\alpha_{ij}$ in Eq. (44).

$$\alpha_{ij} = \frac{\exp(e_{ij})}{\sum_{k=1}^{M} \exp(e_{ik})}. \tag{44}$$

Note that the softmax is applied across temporal dimension M, not across the vector dimension $N_1 + N_2$, as the attention operates between frame-level features. Using the computed attention weights $\alpha_{ij}$, each frame's fused feature vector $f_i$ is reweighted to generate an enhanced representation $f_i'$ Eq. (45).

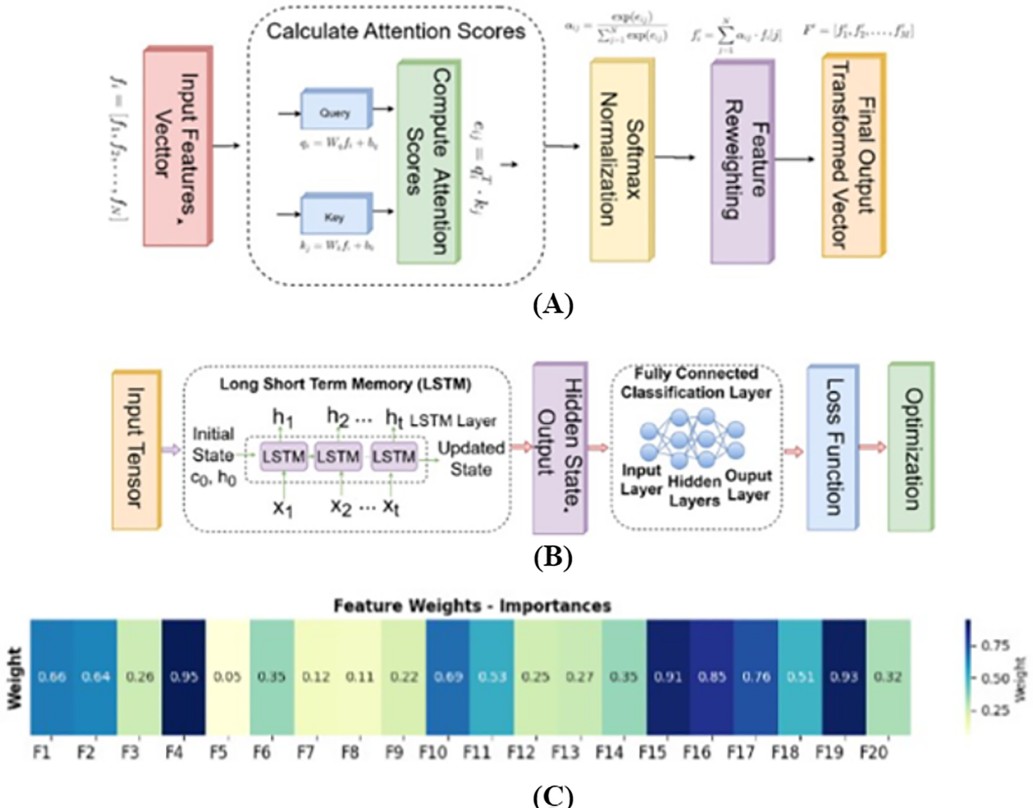

**Figure 6 (A) Architecture of the attention mechanism for feature fusion; (B) design of long short-term memory (LSTM) architecture; (C) heatmap highlighting the feature importances for selected 20 features.**

$$f'_i = \sum_{j=1}^{M} \alpha_{ij} \cdot f_j. \tag{45}$$

This operation aggregates information across the entire sequence, enabling each frame's representation to be informed by context from other frames, which enhances the discriminative power of the features. The final output from the attention module across the video is given in Eq. (46).

$$F' = \left[f'_1, f'_2, \ldots, f'_M\right]. \tag{46}$$

This attention-enhanced sequence $F'$ is subsequently used for downstream tasks such as exercise classification or movement quality assessment. By explicitly defining the learnable parameters $W_q$ $W_k$ and the softmax-based weight assignment logic, this mechanism clarifies how the attention module is integrated into the fusion pipeline. Figure 6A visually represents this architecture, showing how fused features from each frame pass through the attention module to derive a context-aware output that better captures the temporal dynamics and discriminative structure of the input video sequence.

Using an attention mechanism, each feature was assigned a specific weight, indicating its relative importance. Based on these weights, a data-driven selection process was

employed to identify and retain the most relevant features for further analysis or model training. Figure 6C shows the feature importances for the selected features, the features, which exhibit higher weights highlight their greater significance and contribution to the overall task.

## LSTM-BASED CLASSIFICATION

To effectively capture both the spatial and temporal characteristics of sign language gestures, our system integrates frame-level static descriptors with a long short-term memory (LSTM) network for dynamic classification. This approach addresses the inherent complexity of sign language, where both instantaneous hand configurations and their evolution over time are critical for accurate recognition.

Specifically, the keypoint-based descriptors (ORB, AKAZE, SIFT, BRISK) and shape-based features (smoothness, convexity, compactness, symmetry) are extracted independently for each individual frame within a gesture sequence. These descriptors, while static in their per-frame computation, provide a rich representation of the hand's posture and configuration at discrete points in time.

The static descriptors extracted from individual frames are adapted for temporal learning by treating the sequence of frame-level feature vectors as input to an LSTM network. For a dynamic gesture consisting of T frames, the fused keypoint and shape features from each frame $F_t$ are fed sequentially into the LSTM. Leveraging its recurrent architecture, the LSTM captures temporal dependencies by learning movement patterns, transitions, and correlations across successive hand configurations throughout the gesture. This sequential processing enables the model to build a contextual understanding of the gesture's dynamic progression. By analyzing the order and variation in these frame-level features, the LSTM effectively models the temporal evolution of the sign, allowing for accurate classification of gestures defined by motion trajectories rather than static poses. This approach provides a robust mechanism for combining the fine-grained spatial information captured by per-frame static descriptors with the powerful temporal modeling capabilities of LSTMs, thereby justifying their alignment in the context of dynamic sign language recognition.

LSTM networks are particularly suited for sequential data where long-range temporal dependencies are crucial such as in human exercise classification, where subtle changes in motion over time determine the exercise type and correctness. While gated recurrent units (GRUs) offer computational efficiency by simplifying LSTM's gates, LSTMs generally provide better performance in tasks with longer sequences due to their separate memory cell structure. Compared to Transformers, which have shown state-of-the-art performance in various domains, LSTMs require less data and computational resources for effective training and generalize well when the dataset is moderately sized. In our setting, where temporal continuity of skeletal or sensor-based features is key. LSTMs strike a balance between model complexity and learning capacity, making them a practical and effective choice.

An LSTM network classifies exercises as shown in Fig. 6B. LSTMs capture spatial and temporal dependencies, enhancing accuracy in identifying exercise type and correctness.

The transformed feature vectors $f_i'$ are fed into the LSTM as a 3D tensor $F_{input}' \in R^{B \times M \times (N_1 + N_2)}$, where $B$ is the batch size. The stacked LSTM captures temporal dependencies using input $i_t$, forget $f_t$, and output $o_t$, cell state $c_t$ and hidden state $h_t$ updates are defined in Eq. (47).

$$c_t = f_t \odot c_{t-1} + i_t \odot \tanh\left(W_c f_t' + U_c h_{t-1} + b_c\right) \tag{47}$$

This allows the LSTM to capture both short- and long-term temporal dependencies. The LSTM's final hidden state $h_M$ s passed to a fully connected layer, using softmax for multi-class exercise classification or sigmoid for binary correctness evaluation.

## DATASETS DESCRIPTION

This study used the KArSL-100, KArSL-190, and KArSL-502 (*Luqman & Elalfy, 2022*) datasets to enhance ArSL recognition and model generalization. KArSL-100 includes 100 dynamic signs performed by three signers, totaling 15,000 samples. KArSL-190 expands to 190 signs, including digits, letters, and words, with 28,500 samples. KArSL-502, the most extensive dataset, covers 30 digits, 40 letters, and 432 words, totaling 75,300 samples. Each dataset ensures diverse and high-quality sign variations for robust model training.

## EXPERIMENTAL SETUP

To rigorously assess the performance and generalization capability of the proposed system, a comprehensive evaluation protocol was adopted. The KArSL dataset, specifically designed for sign language recognition, includes predefined train-test splits to standardize evaluation across studies. These fixed splits ensure that the training and testing sets are mutually exclusive and representative of real-world conditions. By using the official splits provided with the dataset, we maintain consistency with prior work and avoid potential biases introduced by arbitrary data partitioning. This approach supports a fair and reliable assessment of the system's effectiveness.

## RESULTS AND DISCUSSION

As seen in Table 1, it maintains an average precision, recall, and F1-score of 0.68, indicating strong model reliability with minimal variance. The smaller class set allows for better feature separation, reducing misclassification. For KArSL-190, the accuracy drops to 62.53%, reflecting the increased complexity of the dataset. As shown in Table 1, the precision, recall, and F1-score decrease to 0.50. suggesting greater challenges in differentiating overlapping classes. The presence of visually similar signs contributes to the increased misclassification rate. The KArSL-502 dataset records the lowest performance, with an accuracy of 47.58%. The precision, recall, and F1-score further decline to 0.40. The large number of classes increases the likelihood of feature confusion, leading to a higher misclassification rate.

For, the signer independent mode, the model achieves high classification accuracy on the KARSL-100 dataset, with an average AUC of 0.96, as shown in Fig. 7A. Most classes are well-separated, leading to low false positive rates and strong ROC curves. A few similar signs exhibit minor misclassification, but overall, the model demonstrates excellent

**Table 1 Average precision, recall, F1-score for all datasets (signer independent mode).**

| Dataset | Accuracy | Precision | Recall | F1-score |
|---|---|---|---|---|
| KArSL-100 | 0.7734 | 0.68 | 0.68 | 0.68 |
| KArSL-190 | 0.6253 | 0.50 | 0.50 | 0.50 |
| KArSL-502 | 0.4758 | 0.40 | 0.40 | 0.39 |

performance for a smaller class set. With 190 classes, the dataset introduces higher complexity, leading to a slight decrease in AUC to 0.92, as illustrated in Fig. 7A. While classification remains strong, some visually similar classes show overlapping decision boundaries, resulting in occasional misclassifications. However, the model maintains high recall, ensuring reliable detection across most categories.

As the most complex dataset, KARSL-502 presents challenges due to a large number of classes, resulting in an AUC of 0.87, as depicted in Fig. 7A. Increased class overlap leads to higher false positives, affecting overall precision. While the model still performs competitively, further optimization through improved feature selection or deep learning enhancements could enhance classification accuracy.

For the signer independent mode, the KArSL-100 dataset achieves the highest classification performance, with an accuracy of 77.34%. We used signer independent mode to evaluate the model's generalization ability across different users, which is essential for real-world applications.

The confusion matrices for the KArSL datasets demonstrate strong classification performance. KArSL-100 (Fig. 7B) achieves the highest accuracy, with a nearly perfect diagonal, indicating excellent model performance on this smaller class set. KArSL-190 (Fig. 7B) also exhibits robust classification, with most classes correctly identified and only minor misclassifications occurring between visually similar signs. Despite the increased complexity, KArSL-502 (Fig. 7B) shows a significant accuracy improvement, with slight misclassifications primarily among similar classes. Due to the large number of classes in KArSL-502, it is challenging to display all confusion matrix values clearly, which limits detailed visualization.

Figure 8A illustrates the performance of the KArSL-100 model over 100 epochs. Model accuracy, both training and validation accuracy steadily increase towards approximately 0.8 (80%) and converge, indicating effective learning and good generalization. In Fig. 8B, (Model Loss) mirrors this success, with both training and validation loss rapidly decreasing to near 0.1–0.2 and then flattening out, signifying that the model is successfully minimizing errors without significant overfitting.

For KArSL-190, Fig. 8C depicts the evolution of both training and validation accuracy across 100 epochs. It is evident that both accuracy curves rise steadily, with training accuracy reaching approximately 0.65 and validation accuracy stabilizing around 0.62 by the 100th epoch. This indicates the model is learning, with validation accuracy showing a slight performance gap compared to training accuracy.

In parallel, Fig. 8D illustrates the corresponding training and validation loss values. Both loss curves exhibit a sharp initial decline, followed by a more gradual reduction, ultimately

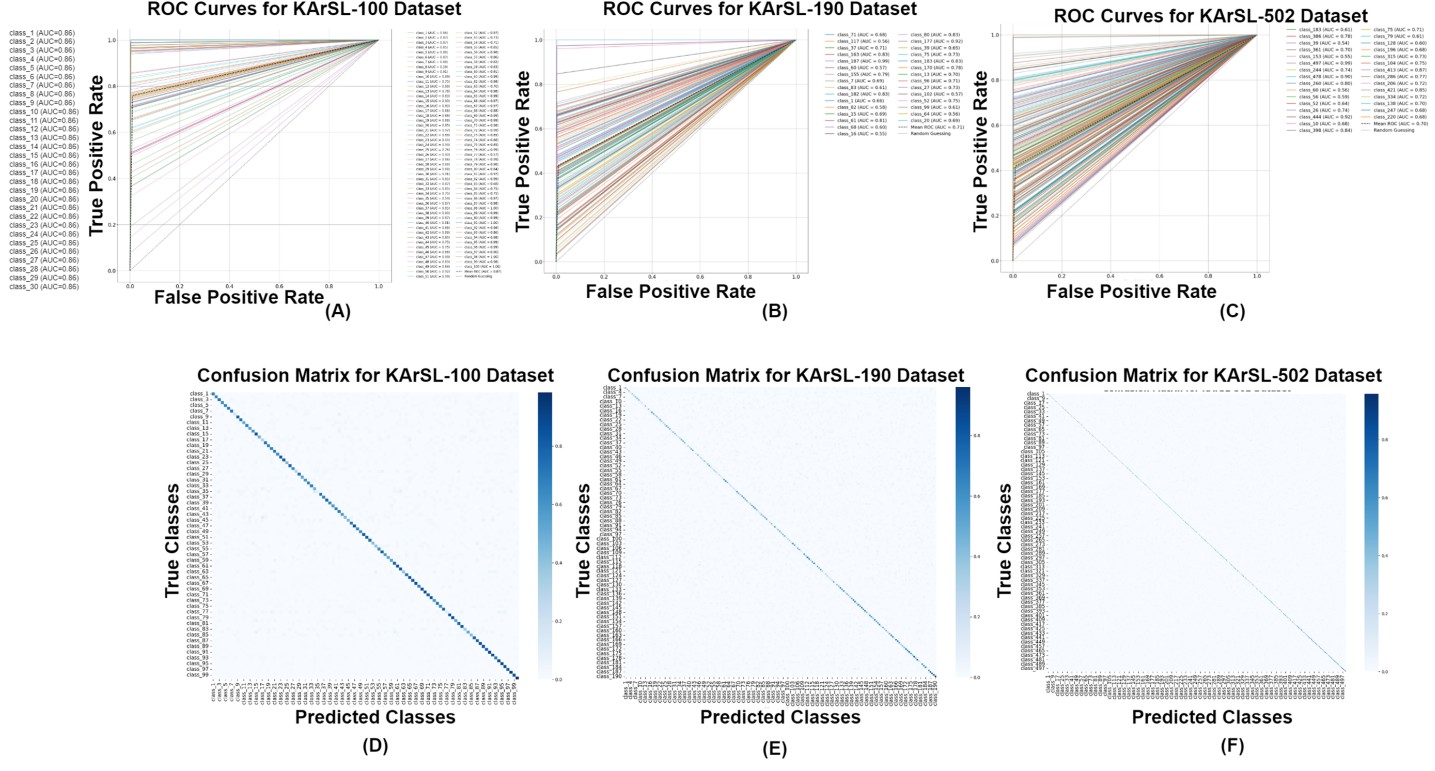

**Figure 7** ROC curves (A) KArSL-100; (B) KArSL-190; (C) KArSL-502 confusion matrix; (D) KArSL-100; (E) KArSL-190; (F) KArSL-502 (selective classes).

reaching near-zero values by the 100th epoch. This consistent decrease in loss confirms the model's effective convergence and error minimization during training. Figures 8E and 8F illustrate the training performance of the KArSL-502 model over 100 epochs. Figure 8E presents model accuracy, where the "Train Accuracy" (blue line) steadily increases from near 0.0 to approximately 0.45, and the "Validation Accuracy" (orange line) similarly rises from near 0.0 to just above 0.4. This upward trend indicates that the model is effectively learning from the data. Figure 8F shows the corresponding loss curves. The "Train Loss" (blue line) decreases from around 1.75 to nearly 0.0, while the "Validation Loss" (orange line) drops from above 1.75 to approximately 0.1. These reductions in loss further confirm the model's learning progress and convergence during training.

## EVALUATION PROTOCOL

Crucially, to validate the system's ability to generalize to new signers and prevent data leakage, a signer-independent evaluation approach was implemented. This ensured that data from any specific signer appeared exclusively in either the training or testing set, preventing the model from learning signer-specific nuances rather than generalizable gesture patterns. Specifically, signers were randomly assigned to folds such that no signer's data contributed to both the training and testing sets within any given fold. 'or' A leave-one-signer-out strategy was considered for robustness.

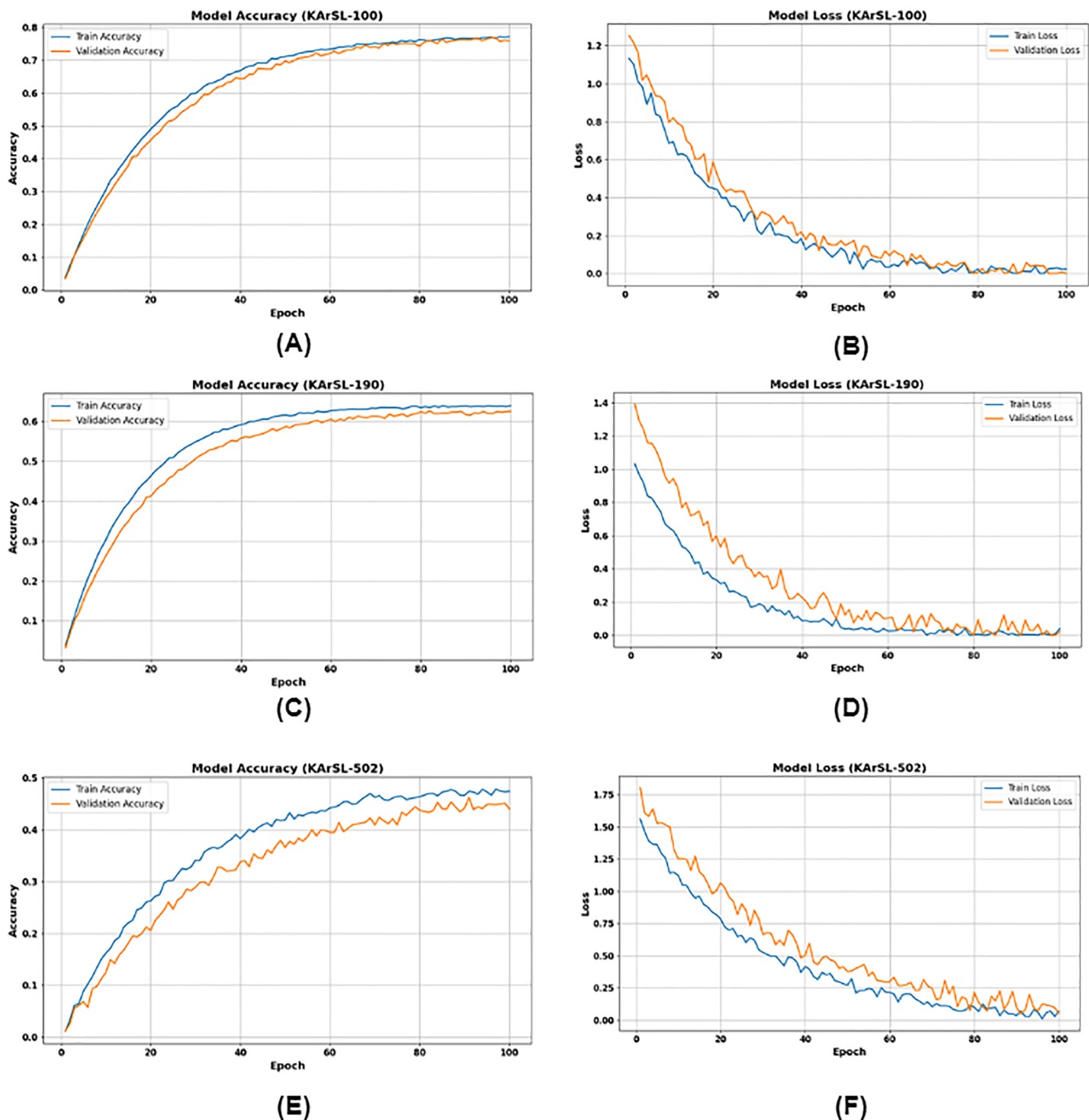

**Figure 8** (A) KArSL-100 model accuracy; (B) KArSL-100 model loss; (C) KArSL-190 model accuracy; (D) KArSL-190 model loss; (E) KArSL-502 model accuracy; (F) KArSL-502 model loss.

To ensure the robustness and statistical significance of our findings, all experiments were conducted over ten independent runs. The results are reported as mean precision, recall, and F1-score to provide a comprehensive view of the model's performance. As shown in Table 1, the model achieved an average accuracy of 77.34% on KArSL-100, with a precision, recall, and F1-score of 0.68. For the larger and more complex KArSL-190 and KArSL-502 datasets, the accuracy declined to 62.53% and 47.58%, respectively, with corresponding drops in precision and recall to 0.50 and 0.40, respectively.

To further validate these results, 95% confidence intervals were calculated. For example, the KArSL-502 accuracy falls within a confidence interval of approximately [46.4%, 48.7%], reinforcing the statistical reliability of our estimates.

A detailed error analysis using confusion matrices and validation accuracy/loss curves highlighted recurring misclassification patterns, particularly in the larger datasets. In KArSL-502, many signs with similar temporal patterns or motion trajectories were frequently confused. Signs involving subtle or brief dynamic transitions also showed higher misclassification rates, suggesting that increased complexity in motion and gesture execution poses challenges for the model.

To mitigate overfitting and promote generalization, we employed early stopping (with a patience of 10 epochs), L2 regularization ($\lambda = 0.001$), and dropout layers (rate = 0.5) in the network architecture. The model's performance on unseen test sets was continuously monitored to ensure stability and robustness.

The observed performance decline across datasets—from KArSL-100 to KArSL-502—can be attributed to the expanding vocabulary, which introduces greater gesture diversity, increased temporal complexity, and a broader range of signer-specific variations. Per-class analysis showed that signs with overlapping movement profiles or brief durations were especially prone to misclassification, contributing significantly to the overall drop in precision, recall, and F1-score in the largest dataset. The time cost analysis is given in Table 2.

## ABLATION STUDY

In Table 3, the performance analysis of the KArSL model across its various configurations highlights the importance of each component in achieving optimal accuracy for sign language recognition. The full model, which integrates preprocessing, keypoints, shape features, attention fusion, and LSTM-based classification, delivers the highest accuracy across all datasets—77.34% for KArSL-100, 62.53% for KArSL-190, and 47.58% for KArSL-502. Omitting the preprocessing stage, which is essential for standardizing input images, leads to a moderate decline in accuracy, emphasizing its role in minimizing noise and enhancing feature extraction. The exclusion of keypoints, which capture fine spatial and motion details using ORB, AKAZE, SIFT, and BRISK, results in the most significant performance drop—underscoring their critical role in distinguishing visually similar signs. Replacing keypoints with distance maps provides a slight improvement over having no keypoints, but still underperforms compared to traditional keypoint-based extraction, suggesting distance maps lack the discriminative power of local descriptors. Lastly, removing shape features and global descriptors of hand geometry also leads to a noticeable

**Table 2 Time cost analysis.**

| Process | Estimated time (s) | MFLOPS |
|---|---|---|
| Preprocessing & silhouette extraction | 0.82 | 52.42 |
| SIFT | 1.45 | 30.04 |
| ORB | 0.31 | 140.54 |
| KAZE | 0.31 | 21.06 |
| BRISK | 0.15 | 35.66 |
| Compactness | 0.15 | 5.28 |
| Symmetry | 1.46 | 0.45 |
| Global convexity | 0.0285 | 13.03 |
| Uniqueness and compactness | 0.0137 | 0.04 |

**Table 3 Ablation study: impact of component removal on ArSL recognition accuracy.**

| Component | Role in the model | Impact when omitted | KArSL-100 accuracy | KArSL-190 accuracy | KArSL-502 accuracy |
|---|---|---|---|---|---|
| Full model | Integrates all features, attention fusion, and LSTM classification. | Baseline performance. | 77.34 | 62.53 | 47.58 |
| Without preprocessing | Prepares hand images/data for feature extraction. | Increases noise and inconsistencies in input, reducing feature extraction accuracy and overall performance. | 72.80 | 58.53 | 42.08 |
| Without keypoints | Extracts detailed spatial and motion hand descriptors (ORB, AKAZE, SIFT, BRISK). | Loses precise spatial information; reduces ability to distinguish visually similar signs based on fine hand movements. | 55.43 | 40.59 | 30.54 |
| Without keypoints using distance map | Extracts features from distance maps of hand silhouettes. | May offer different spatial information, but might miss specific invariant features captured by traditional keypoint detectors. | 61.35 | 52.27 | 34.00 |
| Without shape features | Extracts global hand characteristics (smoothness, convexity, compactness, symmetry). | Weakens representation of overall hand and finger configurations; misses broader context from hand contour and geometry. | 68.30 | 57.40 | 39.35 |

accuracy reduction, indicating their importance in capturing holistic hand configurations. Overall, each component plays a vital role, with keypoints and preprocessing contributing most significantly to model performance.

## COMPARISON WITH STATE-OF-THE METHODS

The classification accuracies across various methods on the KArSL-100 and KArSL-190 datasets highlight the challenges and advancements in ArSL recognition. In signer-dependent scenarios, where models are trained and tested on the same individuals, methods have achieved high accuracies, often exceeding 99%. For instance, *Luqman & Elalfy (2022)* reported a 99.7% accuracy on the KArSL-100 dataset in a signer-dependent setting. However, in signer-independent contexts, where models are evaluated on individuals not seen during training, the performance significantly drops. *Luqman & Elalfy (2022)* reported a 64.4% accuracy on the KArSL-100 dataset in a signer-independent setting. This decline underscores the variability in signing styles among different individuals, which poses a substantial challenge for generalization in sign language

**Table 4 Comparison with other state-of-the-art methods on all datasets.**

| Dataset | Authors | | Accuracy (%) |
|---|---|---|---|
| KArSL-100 | Alamri et al. (2024) | Signer-dependent | 0.9974 |
| | | Signer-independent | 0.682 |
| | Alamri et al. (2024) | Signer-dependent | 0.936 |
| | | Signer-independent | 0.294 |
| | Proposed model | Signer-independent | 77.34 |
| | | Signer-dependent | 97 |
| KArSL-190 | Luqman & Elalfy (2022) | Signer-independent | 0.406 |
| | | Signer-dependent | 0.992 |
| | Luqman & Elalfy (2022) | Signer-independent | 0.402 |
| | | Signer-dependent | 0.991 |
| | Proposed model | Signer-independent | 62.53 |
| | | Signer-dependent | 96.25 |
| KArSL-502 | Alamri et al. (2024) | Signer-independent | 0.273 |
| | | Signer-dependent | 0.989 |
| | Luqman & Elalfy (2022) | Signer-independent | 0.343 |
| | | Signer-dependent | 0.996 |
| | Proposed model | Signer-independent | 47.58 |
| | | Signer-dependent | 93.5 |

recognition systems. The proposed model's performance, with accuracies of 77.34% on KArSL-100 and 62.53% on KArSL-190, indicates progress in developing more robust recognition systems. Nonetheless, the noticeable performance gap between signer-dependent and signer-independent scenarios emphasizes the need for further research to enhance model generalization across diverse signers as given in Table 4.

## LIMITATIONS

Our sign language recognition model encounters challenges due to highly self-occluded body poses, where overlapping limbs obscure key gesture features, and twisted body postures, which distort movement interpretation. Hand gestures near the face create difficulties in distinguishing hand movements from facial expressions, while horizontally aligned hands pose depth perception challenges, making it harder to differentiate between similar gestures. Addressing these limitations is crucial for enhancing the model's accuracy and robustness in real-world scenarios.

## CONCLUSION AND FUTURE RECOMMENDATIONS

This study presented a robust two-hand static and dynamic gesture recognition system for Arabic Sign Language (ArSL), addressing key challenges such as signer variability, occlusions, and intra-class variations. By integrating keypoint-based descriptors (ORB, AKAZE, SIFT, BRISK) with shape-based features (smoothness, convexity, compactness, symmetry) and employing an attention-enabled feature fusion strategy, the proposed method enhances gesture discrimination. Additionally, the use of LSTM networks enables effective modeling of temporal dependencies in dynamic gestures. Experimental

evaluations on KArSL-100, KArSL-190, and KArSL-502 datasets demonstrated 77.34%, 62.53%, and 47.58% accuracy, respectively, showcasing the system's effectiveness across varying dataset complexities. These findings highlight the potential of combining spatial and temporal features for sign language recognition, paving the way for more accurate and inclusive communication technologies for individuals with hearing and speech impairments. Future work will focus on improving generalization to unseen signers, by fusion of full body pose estimzation in existing appoach, and extending the system to real-time applications for enhanced accessibility.

### Funding
This work was supported by the IITP (Institute of Information & Communications Technology Planning & Evaluation)-ICAN (ICT Challenge and Advanced Network of HRD) (IITP-2025-RS-2022-00156326), 50) grant funded by the Korea Government (Ministry of Science and ICT). This work was also supported by the Deanship of Research and Graduate Studies at King Khalid University through Large Group Project under grant number (RGP2/367/46). The funders had no role in study design, data collection and analysis, decision to publish, or preparation of the manuscript.

### Grant Disclosures
The following grant information was disclosed by the authors:
IITP (Institute of Information & Communications Technology Planning & Evaluation)-ICAN (ICT Challenge and Advanced Network of HRD): (IITP-2025-RS2022-00156326), 50), Korea Government (Ministry of Science and ICT).
Deanship of Research and Graduate Studies at King Khalid University through Large Group Project: RGP2/367/46.

### Competing Interests
The authors declare that they have no competing interests.

### Author Contributions
- Zarnab Kausar conceived and designed the experiments, performed the experiments, analyzed the data, performed the computation work, prepared figures and/or tables, authored or reviewed drafts of the article, and approved the final draft.
- Shaheryar Najam conceived and designed the experiments, performed the experiments, analyzed the data, performed the computation work, prepared figures and/or tables, authored or reviewed drafts of the article, and approved the final draft.
- Mohammed Alshehri performed the experiments, authored or reviewed drafts of the article, and approved the final draft.
- Yahya AlQahtani performed the experiments, authored or reviewed drafts of the article, and approved the final draft.
- Abdulmonem Alshahrani analyzed the data, performed the computation work, prepared figures and/or tables, and approved the final draft.

- Ahmad Jalal analyzed the data, performed the computation work, authored or reviewed drafts of the article, and approved the final draft.
- Jeongmin Park analyzed the data, performed the computation work, authored or reviewed drafts of the article, and approved the final draft.

## Data Availability

The code is available in the Supplemental Files.

The KArSL 100, KArSL 190 and KArSL 502 datasets are available at: https://hamzah-luqman.github.io/KArSL/index.html#sec-57f5.

## Supplemental Information

Supplemental information for this article can be found online at http://dx.doi.org/10.7717/peerj-cs.3275#supplemental-information.

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
