# Peer review of "Two-hand static and dynamic Arabic sign language recognition using keypoints and shape descriptors with attention-driven feature fusion"

_PeerJ Computer Science, doi:10.7717/peerj-cs.3275_

## Round 0.1 · original submission · Major Revisions

· Academic Editor

Major Revisions

The problem addressed by the paper is interesting; however the paper needs major revisions related to various aspects, including:
1) related works lack important references to relevant literature;
2) the experimental evaluations require clarifications and integrations
3) the overall presentation would benefit from thorough proofreading.

**Language Note:** The review process has identified that the English language must be improved. PeerJ can provide language editing services - please contact us at [email protected] for pricing (be sure to provide your manuscript number and title). Alternatively, you should make your own arrangements to improve the language quality and provide details in your response letter. – PeerJ Staff

Reviewer 1 ·

Basic reporting

1. The literature review is minimal. Several recent works using attention mechanisms, temporal modeling, or hybrid visual descriptors for sign language recognition are missing.
2. The introduction lacks depth in positioning the contribution with respect to existing multimodal or keypoint-based sign language models.
3. Some abbreviations (e.g., HOG, ORB, LSTM) are not expanded at first use or explained clearly for non-specialist readers.

Experimental design

Fusion mechanism:
1. The attention-driven fusion strategy is mentioned, but the technical explanation is insufficient.
2. There are no equations, no weight allocation logic, and no architectural visualization to clarify how the attention mechanism operates over feature sets.
Dataset splits and validation:
1. The paper does not specify whether cross-validation or a fixed train-test split was used.
2. There is no mention of fold counts, stratification strategy, or if signer-independent splits were used to avoid data leakage in dynamic gestures.
Temporal modeling justification:
1. LSTM is used, but the reason for choosing it over alternatives (e.g., GRU, Transformer) is not provided.
2. Also, the alignment between static descriptors (HOG, LBP) and sequence modeling with LSTM is not clearly justified. It is unclear how static shape descriptors are used in dynamic classification.
3. It is also unclear how static descriptors (like HOG, LBP) are adapted for temporal learning.

Validity of the findings

1. The evaluation lacks statistical testing (e.g., confidence intervals, standard deviations across runs).
2. There is no error analysis, no analysis of per-class performance, and no discussion of overfitting risks.
3. The datasets vary in size and complexity, but there's no explanation on why performance drops, or how specific classes or sign types behave differently.

Additional comments

1. The fusion module would benefit from an ablation study to quantify the contribution of each feature type (keypoint, shape).
2. Attention visualization (e.g., feature weight heatmaps) would help in understanding the interpretability of the model.
3. Comparative discussion can be extended to include hybrid architectures combining CNN and attention-based transformers or sequence models for a more comprehensive positioning.
4. Formatting of some figure captions can be improved for clarity.

Reviewer 2 ·

Basic reporting

The paper addresses an important problem related to computer vision. It proposes an approach for isolated sign language. The paper employs several image processing techniques to extract features from the sign image to be fed into the LSTM model. Additionally, the proposed method has been evaluated on a large dataset. However, the paper has major issues related to the writing, experimental work, and validation. Below are some comments:
- The manuscript needs thorough proofreading to improve clarity and consistency.
- Line 59: The statement "identifying these gestures in images or videos, converting them into words, and translating them into spoken language" is inaccurate. Sign Language Recognition (SLR) maps each sign to its corresponding word in the spoken language—it does not translate entire gestures into spoken sentences.
- Line 71: You've already introduced the abbreviation ArSL for Arabic Sign Language. There is no need to use the full term again. Additionally, it is redefined unnecessarily in line 101.
- Writing Style: The writing is inconsistent and sometimes poor. For example, why are the first letters capitalized unnecessarily in lines 98 and 102?
- Figure 5 Caption: Replace "Gesture 1, 2, and 3" with the actual sign words corresponding to each gesture. This recommendation applies to other figures as well.
- Figures: The quality of the figures is poor and should be improved for better clarity.

Experimental design

- Some of the preprocessing steps might remove important features, such as fingerspelling, which is crucial in sign language recognition.
- Line 211: The values used when converting to YCbCr may not generalize well across different skin tones. How did you address this issue?
- Subsection "Area-Based Thresholding for Hand Extraction": Occlusion detection is mentioned, but there’s no explanation of how your approach handles it. Please elaborate.
- Feature Extraction: What is the total number of features extracted from each frame?
- Line 412: How are the weights of Q and K learned in the self-attention component? Typically, self-attention involves learned parameters, but this isn’t clearly described.

Validity of the findings

- The confusion matrix should not be introduced at the beginning of the experimental section. It belongs in the results analysis section.
- Confusion Matrix Interpretation: The confusion matrices provide little insight. For instance, consider the matrix for the KArSL dataset with 502 classes—it lacks meaningful analysis.
- Table 4: Clarify whether these results correspond to a signer-dependent or signer-independent evaluation protocol. If these results are for signer-independent evaluation, what is the performance per signer? Likewise, if they are signer-dependent, provide signer-wise results.
- What is the inference time of the proposed approach?
- Why are confidence intervals included while the KArSL dataset already provides predefined training and testing splits?
- You should evaluate the performance of the proposed model with each feature type to specify which type of features has the most impact on the final results.
- The paper lacks an error analysis section. This should be added for a comprehensive evaluation.
- Table 4: Use a consistent numbering format for all reported values—either percentages or values between 0 and 1, but not both.

---

## Round 0.2 · accepted · Accept

· Academic Editor

Accept

The paper has been revised according to the reviewers' recommendations.

Reviewer 1 ·

Basic reporting

The introduction and literature review have been strengthened and abbreviations (ORB, SIFT, BRISK, LSTM) are now expanded at first use

Experimental design

All the earlier concerns were addressed properly.

Validity of the findings

1. Statistical testing has been added (confidence intervals, standard deviations across multiple runs).

2. Error analysis and per-class performance analysis are now included, with confusion matrices and validation curves.